# Experimental Study on Basic Mechanical Properties of PVA Fiber-Reinforced Coral Cement-Based Composites

**DOI:** 10.3390/ma16072914

**Published:** 2023-04-06

**Authors:** Jin Yi, Lei Wang, Linjian Ma, Qiancheng Zhang, Jiwang Zhang, Junsheng Chi

**Affiliations:** 1Guangxi Key Laboratory of New Energy and Building Energy Saving, Guilin 541004, China; 2College of Civil Engineering and Architecture, Guilin University of Technology, Guilin 541004, China; 3State Key Laboratory of Disaster Prevention & Mitigation of Explosion & Impact, Army Engineering University of PLA, Nanjing 210007, China

**Keywords:** coral cement-based composites, polyvinyl alcohol fiber, tensile strength, toughen, ductility

## Abstract

In order to improve the brittle characteristics of coral cement-based composites and increase their toughness, an experimental study was carried out on the basic mechanical properties of PVA (polyvinyl alcohol) fiber-reinforced coral cement-based composites, taking into account the fiber content and length-to-diameter ratio (L/D). The results showed that PVA fibers can effectively improve the mechanical properties of concrete, especially its tensile strength. At the same time, PVA fibers improved the damage characteristics of cement-based composites and had obvious toughening and brittleness reduction effects. The PVA fibers, with a volume content of 1.5% and an L/D of 225, had the best performance in reinforcing the overall performance of the coral cement-based composites. Too many PVA fibers or a large length-to-diameter ratio would make it difficult for the fibers to contribute to toughness and cracking resistance and even cause defects in the matrix, reducing the mechanical properties. The tensile stress-strain curves of PVA fiber-reinforced coral cement-based composites were consistent with the trilinear constitutive model curves and showed the tensile characteristic of strain hardening after the occurrence of the main cracks.

## 1. Introduction

There are nearly 10 million square kilometers of coral reefs and coral islands in various forms in waters between 30 degrees north and south latitude on Earth, especially in the central and western Pacific, such as along the Mediterranean Coast, in the Persian Gulf of Iran, and in the Red Sea of Egypt. Coral reefs are special rock masses formed by the limestone shells of reef-building coral colonies over long periods of compaction and fossilization after they die, in which the colonies secrete calcium carbonate and absorb carbon dioxide from seawater as they grow. Sometimes, under the action of hydrodynamics, coral reefs are broken down into coral sand with calcium carbonate as the main component, which is a typical marine rock and soil with a wide particle size distribution, mainly distributed in sandy and silty soil sequences [1]. Marine engineering constructions far from the mainland are restricted by their geographical location and ocean conditions, so transporting building materials from the mainland on a grand scale is neither economical nor practical. The local use of coral sand as an aggregate to prepare cement-based materials can effectively solve the problem of the lack of traditional building materials, which has outstanding economic benefits and strategic significance [2,3,4].

Coral sand is classified as a natural light aggregate due to its light weight, porosity, and low cylindrical compressive strength. Its particles are irregular in shape and prone to breakage, and it has abundant pores its surface and inside. The physical properties of coral sand are completely different from those of terrestrial sand [5,6]. Therefore, the porosity of the coral concrete prepared by conventional methods is as high as approximately 20%, which is obviously higher than that of ordinary sand concrete. Further improvement of the strength grade of coral concrete when it reaches C30–C50 is difficult [7,8,9]. Under uniaxial compression, coral concrete has a steeper stress-strain curve, which is typical of splitting failure. The higher the strength grade of coral concrete is, the stronger the brittleness character [10,11]. Coral concrete has great discreteness in mechanical characteristics, and its axial compression ratio and tensile compression ratio are higher than those of ordinary sand concrete [12,13]. The elastic modulus of coral concrete is lower than that of ordinary sand concrete and lightweight aggregate concrete with the same strength grade, and Poisson’s ratio is similar to that of lightweight aggregate concrete with the same strength grade [14]. The above studies show that compared with ordinary sand cement-based composites, coral cement-based composites have disadvantages such as low strength, high brittleness, and large deformation, which have hindered the extensive use of coral concrete [15,16], especially for protective engineering.

Since the 1960s, fiber reinforcement and material composites have become important topics in construction materials research. Romaldi et al. [17] suggested improving the brittleness of concrete and limiting the development of cracks by adding fibers, and the tensile strength and toughness of concrete were greatly enhanced. Rao Lan et al. [18] proposed that PVA fiber-reinforced coral concrete has the characteristics of high ductility, high toughness, and multiple cracking, which can effectively address the weaknesses of the brittle failure of concrete. Takashif et al. [19] found that, under a high stress level, cement-based composites have better resistance to fatigue damage and stronger resistance to deformation and multipoint cracking than ordinary fiber-reinforced concrete. Kong Yan et al. [20] showed that the tensile strength and flexural strength of engineering cement-based composites mainly depend on the fiber volume content, and the fiber content controls the strain hardening and softening behavior of the engineering cement-based composites. Izaguirre et al. [21] prepared polypropylene fiber lime-based mortar. Polypropylene fibers can not only improve the tensile properties of lime-based mortar but also reduce its plastic shrinkage and early drying shrinkage deformation. PVA fibers can restrain crack expansion in cement-based composites, improve the mechanical properties of cement-based composites, enhance toughness, and prolong the fatigue fracture life [22,23].

Due to the particularity of coral cement-based composites, the large amount of salt, and the relatively limited application scope, existing studies on fiber-reinforced coral cement-based composites are mainly aimed at those with strength grades below C50 and are still in the initial exploration stage [24,25]. To meet the requirements on coral cement-based composites for blast resistance and impact resistance in offshore reef protection projects and change the current situation that coral cement-based composites can only meet the needs of civil use and some special cases, in this study, based on the low-cost preparation technology of medium- and high-strength coral cement-based composites in the early stage [26], PVA fibers with high toughness, high crack resistance, high deformation, and corrosion resistance, which are used in common cement-based composites, are adopted to reduce the brittleness and improve the toughness of high-strength coral cement-based composites [27]. The basic mechanical properties of PVA fiber-reinforced coral cement-based composites with high strength were tested, and the effects of the PVA fiber content and length-to-diameter ratio (L/D) on the mechanical properties of the coral cement-based composites were analyzed to reveal the strengthening mechanism of its PVA fibers. The research results can lay a theoretical foundation for the preparation of high-strength and high-toughness coral cement-based composites and their application in protective engineering.

## 2. Experimental Scheme

### 2.1. Raw Materials and Basic Properties

The aggregate was natural coral sand dredged from the Guangxi Beibu Gulf, and its main chemical component was calcium carbonate. After refining the particle size of coral sand and optimizing particle gradation and sieving, medium sand was prepared with a fineness modulus of 2.68, and the gradation curve is shown in Figure 1, according to Sand for Construction (GB/T 14684-2022). Ordinary Portland cement (P.O. 52.5), fly ash (class I), and quartz powder (average particle diameter: 23 μm, density: 2.72 g/cm^3^) were used in the tested mixtures. The mixing water was artificial seawater prepared according to the composition of the seawater in the South China Sea. The admixture was a polycarboxylate water reducer with a 20% water-reducing rate. In addition, for the desired dispersion of fibers in the mixtures, a polyether-modified polysiloxane defoamer and a hydroxyethyl cellulose thickener were also used in this study. PVA fibers from Kuraray Co., Ltd., Tokyo, Japan, were used as an external material in this test. The performance parameters are shown in Table 1.

### 2.2. Experimental Mix Design and Preparation Method

With the content and L/D of PVA fibers as variables, a total of 9 groups of PVA fiber-reinforced coral cement-based composites were designed based on the benchmark mix proportion, and the water-cement ratio was 0.24. Coral cement-based composites without fibers were used as the control group (NF group). The mixtures were numbered according to their content and length of fibers. For example, PVA-L06-V1.0 has a PVA fiber volume content of 1.0% and a length of 6 mm. The mixed proportions of the samples are shown in Table 2. The size of cubic compressive strength specimens was 100 mm × 100 mm × 100 mm; the size of axial compressive strength specimens and prismatic flexural specimens was 100 mm × 100 mm × 400 mm; and the size of thin-slab specimens for bending tests was 15 mm × 100 mm × 400 mm. The dog-bone specimens recommended by the Japan Society of Civil Engineers (JSCE 2008) were used as the tensile specimens. Their dimensions are shown in Figure 2a. Form stripping of the specimens was carried out 24 h after pouring, and they were placed in an environment of (20 ± 2) °C and 95% humidity for curing until 28 days later. Then, various mechanical property tests were carried out.

### 2.3. Liquidity Test

According to the Concrete Quality Control Standard (GB50164-92), a standard slump tube (upper diameter: 100 mm, lower diameter: 200 mm, height: 300 mm) was used to test the slump of each group of fresh coral cement-based composites. The coral cement-based composites without fibers in the NF group had the best fluidity and excellent construction performance. When PVA fibers were added, the fluidity loss was positively correlated with the fiber volume content and L/D in the fresh coral cement-based composites, as shown in Figure 3. When the volume content of PVA fibers increased to 2% and the L/D increased to 300 (L = 12 mm), the fluidity of the coral cement-based composites was almost completely lost. That is, PVA-L12-V2.0 had the lowest slump (only 33 mm), which belongs to low-plastic cement-based materials. The slump of other fresh PVA fiber-reinforced coral cement-based composites was between 80 mm and 190 mm with good fluidity. This was because PVA fibers have good hydrophilicity; with increasing fiber content, more cement slurry was adsorbed on the surface of the PVA fibers, resulting in the mixture becoming sticky and the fluidity decreasing. In a certain range, increasing the L/D of PVA fibers was beneficial for the fibers to play a bridging role in fresh coral cement-based composites and restrain the flow. However, the Code for Construction of Concrete Structures (GB50666-2011) requires that when the pouring environment temperature is higher than 35℃, the slump should not be less than 70 mm. Therefore, this study suggests that in PVA fiber-reinforced coral cement-based composites, the PVA fiber volume content should not exceed 1.6% and the L/D should not exceed 240 in engineering construction in hot and humid ocean environments.

### 2.4. Test Method for Mechanical Properties

According to the Concrete Physical and Mechanical Properties Test Method Standard (GBT50081-2019), an electronic universal testing machine was used to test the mechanical properties of cubic compression specimens, prismatic compression specimens, prismatic flexural specimens, and thin-slab bending specimens. A multiangle rotating custom chuck was used to fix the tensile specimens in the center, and tensile tests were carried out with loading at both ends. The effective crack observation area was the 80-mm-wide area in the middle of the dog-bone specimens, and the deformation within the 80 mm scale distance was measured by a displacement extensometer. The dog-bone tensile specimen and test loading device are shown in Figure 2.

## 3. Experimental Phenomenon and Failure Modes

Observing the experimental phenomena, the cubic compression specimens and prismatic compression specimens in the NF group had no obvious signs before cracking, The central matrix cracked during the failure, with an angle of approximately 30° between the main crack and the loading direction. Large cement-based caving caused the overall failure, as shown in Figure 4a and Figure 5a. The flexural specimens, thin-slab bending specimens, and tensile specimens in the NF group all rapidly fractured after loading, showing typical brittle failure characteristics, as shown in Figure 6a, Figure 7a, and Figure 8a. Their sections were relatively flat, and a coral-sand penetrating fracture was clearly visible on the sections.

When PVA fibers were added, the vast majority of the coral sand on the section still fractured. However, the failure patterns of the coral cement-based composites significantly changed, and the failure process was relatively slow. The PVA fibers reduced the brittleness of the coral cement-based composites to a certain extent. Some obvious fine cracks appeared when PVA-fiber-reinforced coral cement-based composite failure occurred under compression. The angle between the main cracks and loading direction decreased; the main crack direction of most prismatic specimens was roughly parallel to the loading direction; the failure process was relatively slow; little cement-based caving occurred; and the specimens still maintained good integrity, as shown in Figure 4b and Figure 5b. Under tensile stresses, cracks occurred on the surface and inside of the PVA fiber-reinforced coral cement-based composite flexural specimens and thin-slab bending specimens from bottom to top, and on the section, PVA fibers were pulled out or broken to different degrees, as shown in Figure 6b and Figure 7b. Transverse cracks occurred in the PVA fiber-reinforced coral cement-based composite tensile specimens under tensile stresses. PVA fibers across the cracks exerted their tensile properties through sliding friction and a pull-out fracture. Uneven PVA fibers can be seen to be pulled out or broken on the section, as shown in Figure 8b.

## 4. Discussion of Test Results

### 4.1. Compressive Performance

Due to the many defects of coral sand itself, the compressive strength of coral cement-based composites prepared by conventional methods is generally between C20 and C40 [11]. There is a linear relationship between the axial compressive strength (*f*_c_) and the cubic compressive strength (*f*_cu_), which is described by the linear model *f*_c_ = *βf*_cu_. β is higher than that of ordinary cement materials, which is between 0.76 and 0.95 [6,11,28]. The cubic compressive strength (*f*_cu_) and axial compressive strength (*f*_c_) of the coral cement-based composites prepared in this study were 61.8 MPa and 53.8 MPa, respectively. After the addition of PVA fibers, there was a similar linear relationship between *f*_c_ and *f*_cu_, with *β* = 0.81~0.97.

In general, as shown in Figure 9, the addition of PVA fibers had no significant enhancement effect on the compressive strength of coral cement-based composites in this study. In particular, PVA fibers with an L/D of 225 (L = 9 mm) had the best enhancement effect on the compressive strength of coral cement-based composites. When the PVA fiber volume content was 1.5%, the compressive strength increased to its maximum, *f*_cu_ increased by 11%, and *f*_c_ increased by 23%. This was because the randomly distributed PVA fibers could restrain the transverse expansion of coral cement-based composites under unidirectional pressure [29] and inhibit the generation and development of cracks, thus improving the compressive strength and significantly improving the overall failure characteristics of the specimens. When PVA fibers with an L/D of 150 (L = 6 mm) were added, the compressive strength of the coral cement-based composites decreased with increasing PVA fiber content. Compared with the NF group, when the PVA fiber volume content was 1%, 1.5%, and 2%, *f*_cu_ decreased by 20%, 17%, and 15%, and *f*_c_ decreased by 13%, 10%, and 7%, respectively. The main reason was that the compressive strength of PVA fiber-reinforced coral cement-based composites is mainly provided by the coral cement matrix, and loose and porous coral sand contains more air inside. In the process of mixing and absorbing water, some air can be expelled from the slurry. However, PVA fibers mixed in slurry form a complex fiber-space network that prevents air from being expelled from the matrix. Air that has not been expelled formed pores between the cement paste and coral sand after matrix hardening, which caused an increase in the porosity of the coral sand in the hardened coral cement matrix and increased internal defects. At the same time, it would weaken the interface between the fibers and coral cement matrix, affect the bonding performance between them, and reduce the matrix's compressive strength and compactness. Therefore, the influence of PVA fibers on the coral cement-based composite’s compressive strength was both positive and negative.

Based on an analysis of the test results, there was a certain rule that the compressive strength of PVA fiber-reinforced coral cement-based composites should show an obvious decline-rise-decline with an increased fiber L/D. The effect of the PVA fiber L/D on coral cement-based composites was obvious. This was because when the length of PVA fibers was too short, the restraint effect in resisting coral cement-based composite damage was weak, but internal defects of the matrix were added, which reduced the compressive strength. When the length of the PVA fibers was too long, they easily curled and intertwined with each other, and it was difficult for them to play a reinforcement role, causing the compressive strength of the coral cement-based composites to decrease further. Therefore, the L/D of PVA fibers was an important factor affecting the compressive strength reinforcement effect of coral cement-based composites.

### 4.2. Bending Performance

Combining the failure process in the bending tests of PVA fiber-reinforced coral cement-based composite thin slabs and the load-deflection curve characteristics, as shown in Figure 10, the analysis showed that there were two peaks in the load-deflection curves, similar to those of fiber-reinforced ordinary sand cement-based composites [22], which could be roughly divided into the following four stages:Elastic—rising stage: The load-deflection curves showed a good linear relationship, the coral cement matrix and PVA fibers shared the external force, and the load proportion of the coral cement matrix was greater than that of the fiber;Cracking—declining stage: When the load reached its first peak value, cracks began to appear at the bottom of the specimens. At this time, the coral cement matrix at the cracks was no longer functional, and the PVA fibers had to bear the load. With the development of cracks, the bearing capacity of the specimen decreased;Cracking—picking up stage: PVA fibers across cracks gradually played a bridging role, which transferred the stress to the uncracked coral cement matrix wrapped around them, effectively preventing the continuous decline of the bearing capacity. As the load slowly increased, the stress transmitted by the PVA fibers gradually increased, the coral cement matrix produced new cracks, and the load-deflection curves showed a small fluctuation rising trend;Hardening stage: After the load reached the second peak, the fine cracks in the bottom of the thin slabs grew and gradually formed main cracks, the coral cement matrix lost its bearing capacity, and the bonds between the PVA fibers and matrix were not sufficient to resist the tensile stress borne by the PVA fibers. The PVA fibers across the cracks were broken or pulled out, the load-deflection curves decreased, and the main cracks continued to extend to the top of the specimen until failure.

PVA fibers have good crack resistance and high tensile strength. Theoretically, increasing the fiber length and content is beneficial for improving cement strength. The fibers can share more stress with the coral cement matrix, consume more energy, enhance the ultimate tensile strain of coral cement-based composites, and play a positive role in promoting the thin-slab bending strength and flexural strength of coral cement-based composites. The test results showed that PVA fibers can greatly improve the bending strength of coral cement-based composites, and the bending strength of these composites enhanced by PVA fibers was 125–279% higher than that of the NF group, as shown in Figure 11. The flexural strength also improved, as shown in Figure 12. The flexural strengths of PVA-L09-V1.0 and PVA-L09-V1.5 had the largest increases of 66% and 70%, respectively. In general, the bending strength and flexural strength of coral cement-based composites first increased and then decreased with increasing PVA volume content and L/D. This was because the extent of improvement of the coral cement matrix toughness was greatly affected by the fiber properties, content, and interfacial properties, and the interfacial properties of fibers depended on their strength, toughness, surface condition, dispersibility, and other factors [30]. The PVA fibers had hydrophilic properties [31], and more hydration products adhered to their surface (as shown in Figure 13), which made the smooth PVA fibers have a good bonding effect with the coral cement matrix. However, PVA fibers tend to curl when they are too long, and it is difficult for them to exert their tensile strain capacity when cracks occur in the bridging matrix, which limits the improvement of the coral cement-based composite ductility [32,33]. Too high a PVA fiber content would decrease the fluidity of fresh coral cement-based composites, and then PVA would be unevenly dispersed, leading to more fiber assemblies and bundles in the coral cement-based matrix. The fibers could not fully bond with the matrix, and the high elastic modulus and high tensile strength of the fibers would be underutilized. The fibers would have difficulty playing the roles of toughening and crack resistance, even causing defects in the matrix and reducing the mechanical properties of the matrix. Therefore, too many PVA fibers would limit or further weaken their toughening effect.

According to the analysis of the test results, the bending-compressive strength ratio of PVA fiber-reinforced coral cement-based composites in this study varied from 0.082 to 0.109 with fiber content and L/D. As shown in Figure 14, the bending-compressive strength ratio of PVA fiber-reinforced coral cement-based composites with PVA fiber contents of 1.0% and 1.5% first increased and then decreased with L/D. PVA-L09-V1.0 and PVA-L09-V1.5 had the largest growth rates of 54% and 53%, respectively. However, that of the specimens with a 2.0% PVA fiber volume content continuously decreased with increasing fiber L/D. This was because the PVA fibers had no obvious enhancement effect on the compressive strength, but when the content of PVA fibers was too high, the increase in the flexural strength of coral cement-based specimens was smaller than that of the compressive strength. Therefore, the bending-compressive strength ratio of PVA fiber-reinforced coral cement-based composites with a high content of PVA fibers decreased with increasing L/D. In general, the toughness and crack resistance of PVA fiber-reinforced coral cement-based composites were always better than those of its plain composites, indicating that the beneficial effects of PVA fibers were always greater than the adverse effects.

### 4.3. Tensile Performance

The tensile stress-strain curves of PVA fiber-reinforced coral cement-based composites measured in the tests are shown in Figure 15, and their characteristics were highly consistent with the trilinear constitutive model [34], as shown in Figure 16. Here, the trilinear constitutive relation is:(1)$$\sigma (\varepsilon )=\left\{\begin{array}{c} E_{t}.\varepsilon ,\varepsilon \leq \varepsilon_{t}\\ \sigma_{t}+E_{2}(\varepsilon -\varepsilon_{t}),\varepsilon_{t}<\varepsilon \leq \varepsilon_{sc}\\ \sigma_{sc}+E_{3}(\varepsilon -\varepsilon_{sc}),\varepsilon_{sc}<\varepsilon \leq \varepsilon_{\mathrm{tu}} \end{array}\right.$$
where $E_{t}$ is the initial elastic modulus (OA segment); $E_{2}$ is the elastic modulus in the stable cracking stage (AB segment); $E_{3}$ is the elastic modulus in the crack propagation stage (BC segment); $\sigma_{t}$ is the stress at the initial crack point (point A); $\sigma_{\mathrm{sc}}$ is the stress at the crack saturation point (point B); and $\varepsilon_{\mathrm{tu}}$ is the strain at the ultimate tensile point (point C).

The tensile stress-strain curves of PVA fiber-reinforced coral cement-based composites showed three stages as follows:Elastic rising stage: This stage could be divided into a former stage and a latter stage. That is, in the early stage of the elastic stage, the slope of the stress-strain curves was large, and the strain increase was small. At this time, the stress was mainly borne by the coral cement matrix, which was the former elastic stage. As the tensile stress increased, the PVA fibers in the coral-cement matrix gradually shared a small part of the stress, and the slope of the stress-strain curve became slightly smaller, which was more obvious when the length of PVA fibers was 9 mm and 12 mm, and there was no obvious crack in this stage, which was the latter elastic stage. Therefore, increasing the fiber length within a certain range could improve the early ductility of coral cement-based composites. When the PVA fiber length was 6 mm, the slope of the tensile stress-strain curve (i.e., the initial elastic modulus) increased with increasing PVA fiber content, but with increasing fiber length, the initial elastic modulus of the coral cement-based composites showed little difference. A short length of PVA fibers was beneficial for utilizing their advantages of high elastic modulus and improving the large deformation characteristic of coral cement-based composites to a certain extent;Plastic deformation stage: When the first crack appeared on the relatively weak part of the specimen surface, an obvious turning point appeared in the rising stage of the stress-strain curves. The PVA fiber-reinforced coral cement-based composites entered the plastic deformation stage, the load at the crack was borne by the PVA fibers, and the fibers began to play a bridging role. Then, small cracks gradually increased; with increasing strain, the stress remained relatively stable, and the curves showed a horizontal state in which the curves fluctuated slightly up and down;Failure stage: The section at the cracking site decreased, a subtle crack gradually developed into a main crack, and the stress-strain curves became smooth and rapidly rose. When the load reached the highest point, the PVA fibers were gradually broken or pulled out, the stress-strain curves decreased, and the specimens were fractured.

The medium- and high-strength coral cement-based composites prepared in this study were brittle; their ultimate tensile strain was only 2.5%, and their tensile strength was only 0.85 MPa, which was approximately 30% of the tensile strength of ordinary sand-coral cement-based composites with the same strength grade. When PVA fibers were added to the coral cement-based composites, they formed a random distribution of space grid structures. In coordination with the wrapped coral cement matrix, under uniaxial tensile stress, the coral cement-based composite’s tensile strength could be improved. Even if the coral cement matrix cracked, the fibers would not break immediately and hold the coral cement-based composite specimens such that the tensile strength was not quickly lost, showing excellent strain-hardening tensile properties. In this study, the ultimate tensile strain of most specimens increased to approximately 5–10%, which was an increase of 61–292% compared to that of NF group specimens, as shown in Figure 17. At the same time, the tensile strength of PVA fiber-reinforced coral cement-based composites was also greatly improved to 1.4~7.4 times the tensile strength of NF group specimens, as shown in Figure 18. Generally, the tensile strength increased with increasing PVA content and L/D within a certain range, but when the PVA fiber content was too high or the L/D was too high, the reinforcement effect on the tensile strength was limited or weakened. In this study, the coral cement-based composite specimens with a 1.5% PVA volume content and 225 L/D (L = 9 mm) had the best tensile toughness, and the tensile strength was 638% higher than that of the NF specimens.

## 5. Conclusions

The experimental investigation on the macroscopic mechanical properties and microstructure of the PVA concrete was carried out to reveal the mechanism by which PVA improved the mechanical properties of the concrete. Furthermore, the influence of PVA content and L/D on the mechanical properties of PVA concrete is explored.

Considering that the fluidity of fresh PVA fiber-reinforced coral cement-based composites is significantly affected by the fiber content and L/D ratio, this paper suggests that the fiber volume content of PVA fiber-reinforced coral cement-based composites should not exceed 1.6% and that the length-diameter ratio should not exceed 240 in engineering construction in hot and humid ocean environments;The compressive strength, flexural strength, bending strength, and tensile strength of the coral cement-based composites can be improved by adding different volume contents and L/D ratios of PVA fibers, among which the effect on enhancing the tensile strength was the most significant. At the same time, PVA fibers improved the damage characteristics of the coral cement matrix and had obvious toughening and brittleness reduction effects. The PVA fibers with a volume content of 1.5% and an L/D of 225 have the best performance regarding the overall performance of reinforced coral cement-based composites;PVA fibers tend to curl when they are too long, and it is difficult for them to exert their tensile strain capacity when cracks occur in the bridge’s matrix, which limits the improvement of the coral cement-based composite’s ductility. When the content of PVA fibers is too high, the dispersion is poor in the coral cement matrix, leading to failure to fully bond with the matrix. The fibers have difficulty playing the roles of toughening and crack resistance and even cause defects in the matrix, reducing the mechanical properties of the matrix;There are two peaks in the bending load-deflection curves of PVA fiber-reinforced coral cement-based composites, which are similar to those of fiber-reinforced ordinary sand cement-based composites. The curves can be roughly divided into four stages: elastic—rising stage; cracking—declining stage; cracking—picking up stage; and hardening stage;The tensile stress-strain curves of PVA fiber-reinforced coral cement-based composites can be divided into three stages: the elastic rising stage, plastic deformation stage, and failure stage, which is highly consistent with the trilinear constitutive model. PVA fibers can maintain the tensile strength of coral cement-based composite specimens, which is not rapidly lost after the main crack occurs, showing the tensile property of strain hardening, and the ultimate tensile strain increases by 61~292%.

## Figures and Tables

**Figure 1 materials-16-02914-f001:**
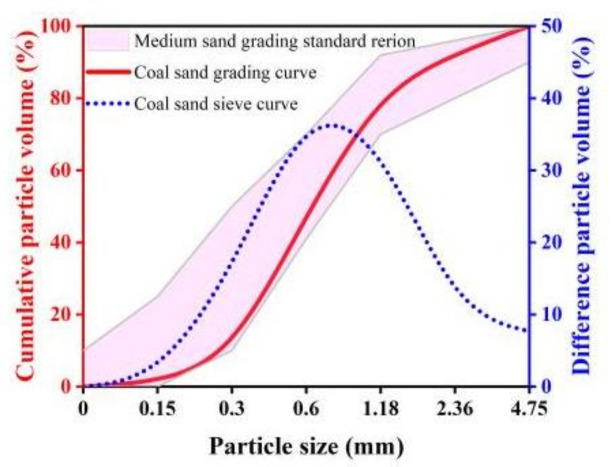
Particle size distribution curve.

**Figure 2 materials-16-02914-f002:**
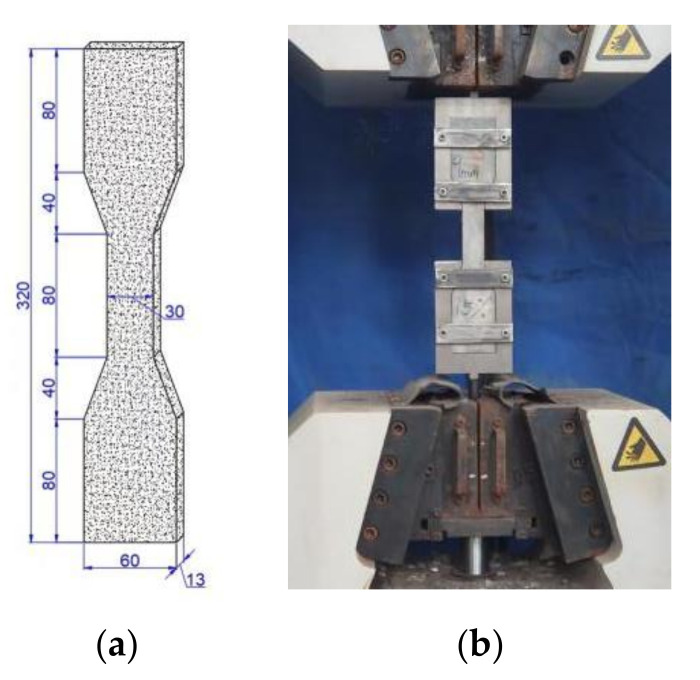
Tensile specimen and tensile loading device. (**a**) Dog-bone specimen (unit: mm); (**b**) Tensile loading device.

**Figure 3 materials-16-02914-f003:**
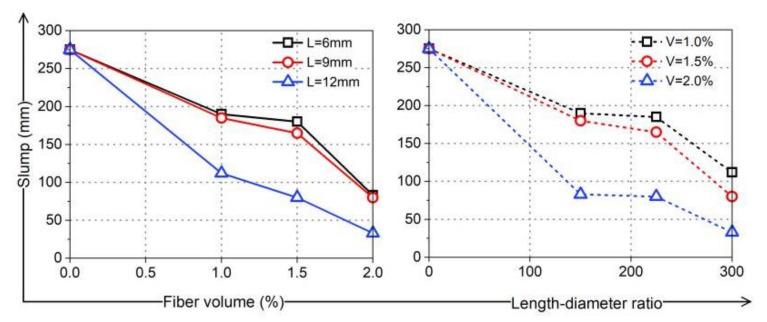
The slump of fresh coral cement-based composites.

**Figure 4 materials-16-02914-f004:**
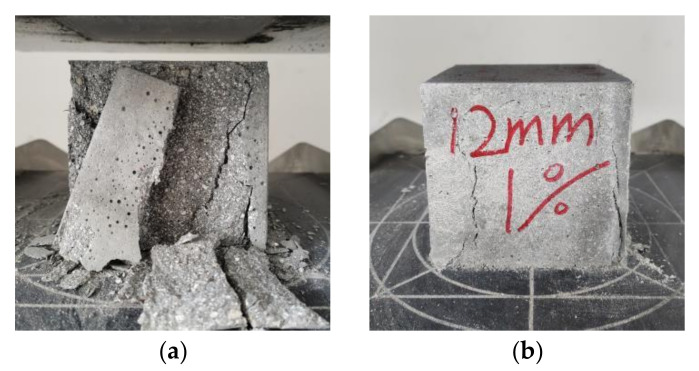
Failure patterns of cubic compression specimens. (**a**) NF, (**b**) PVA.

**Figure 5 materials-16-02914-f005:**
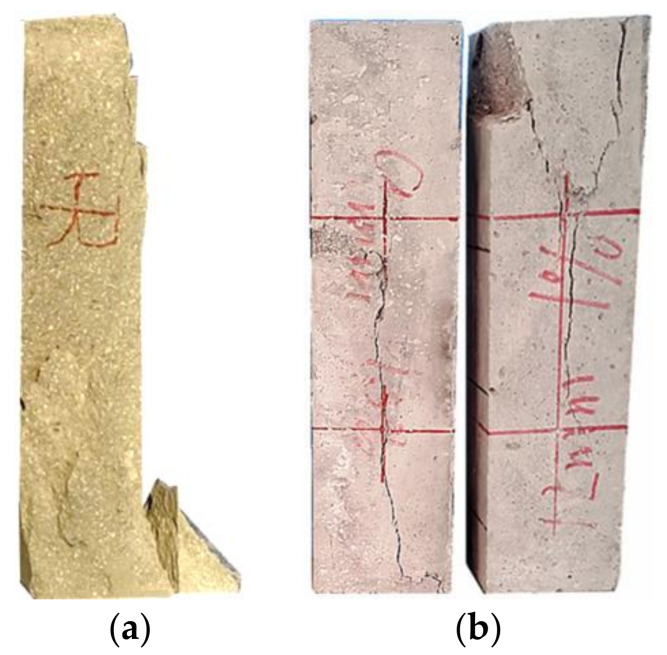
Failure patterns of axial compression specimens. (**a**) NF, (**b**) PVA.

**Figure 6 materials-16-02914-f006:**
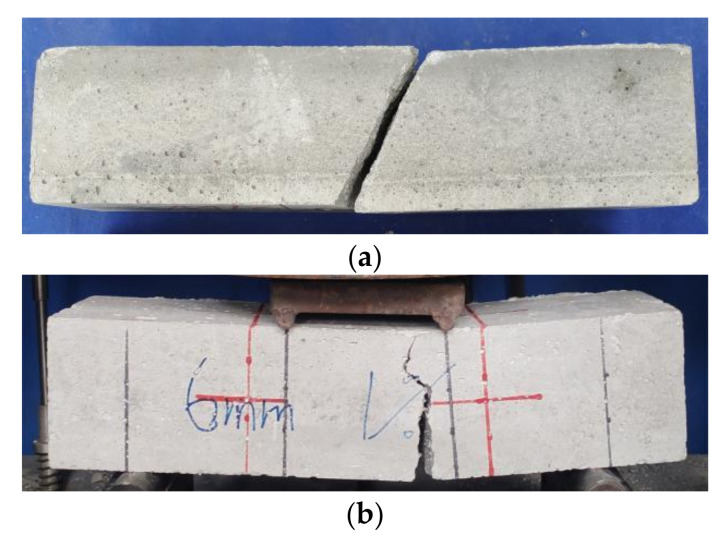
Flexural failure pattern of prismatic specimens. (**a**) NF, (**b**) PVA.

**Figure 7 materials-16-02914-f007:**
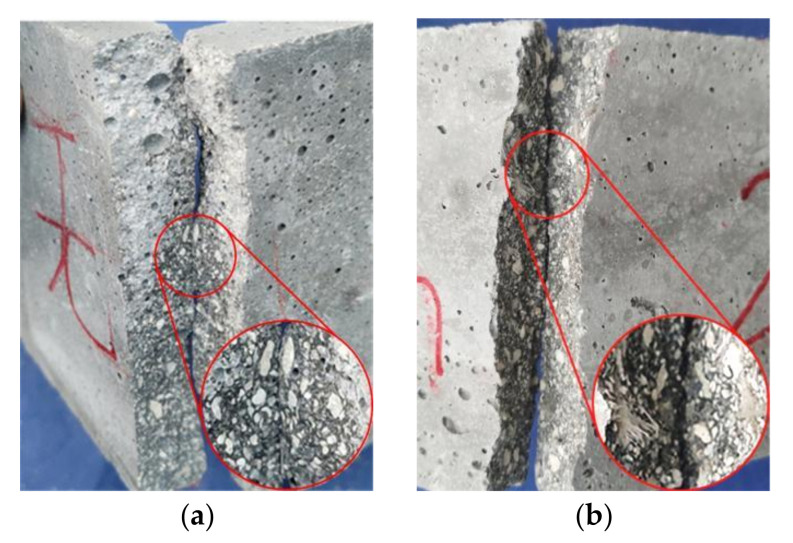
**Figure 7.** Four-point bending failure pattern of thin-slab specimens. (**a**) NF, (**b**) PVA.

**Figure 8 materials-16-02914-f008:**
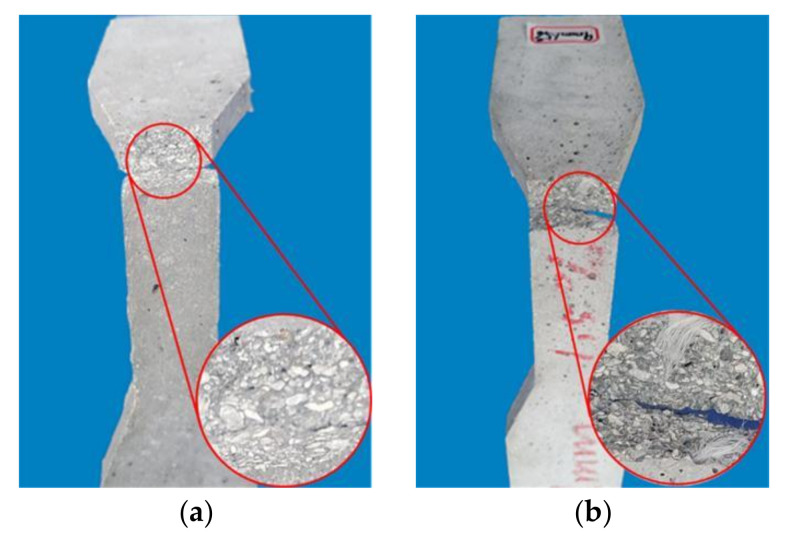
Tensile failure pattern of dog-bone specimens. (**a**) NF, (**b**) PVA.

**Figure 9 materials-16-02914-f009:**
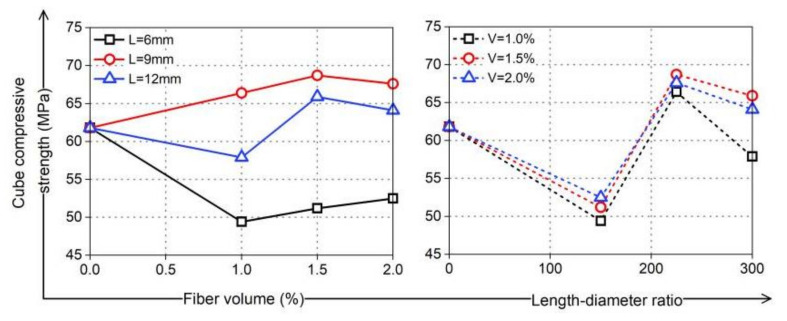
Cubic compressive strength of PVA-reinforced coral cement-based composites.

**Figure 10 materials-16-02914-f010:**
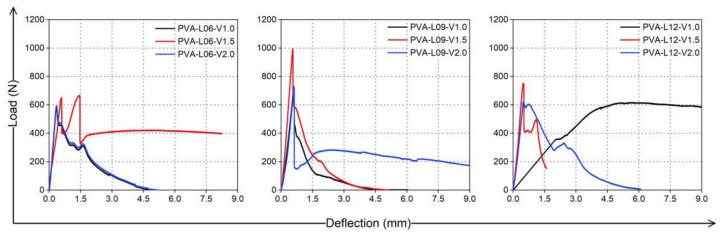
Full load-deflection curves of PVA fiber-reinforced coral cement-based composites specimens in the bending test.

**Figure 11 materials-16-02914-f011:**
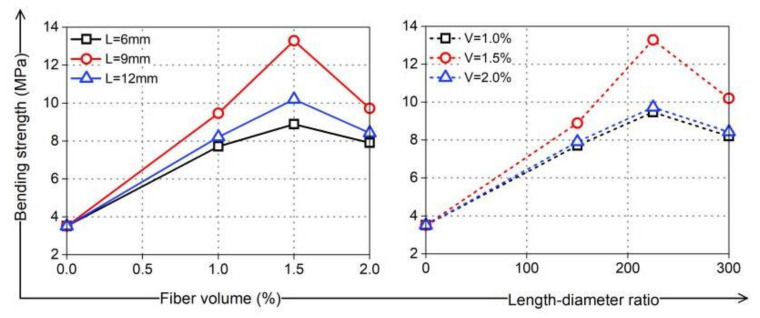
Bending strength of PVA-reinforced coral cement-based composites.

**Figure 12 materials-16-02914-f012:**
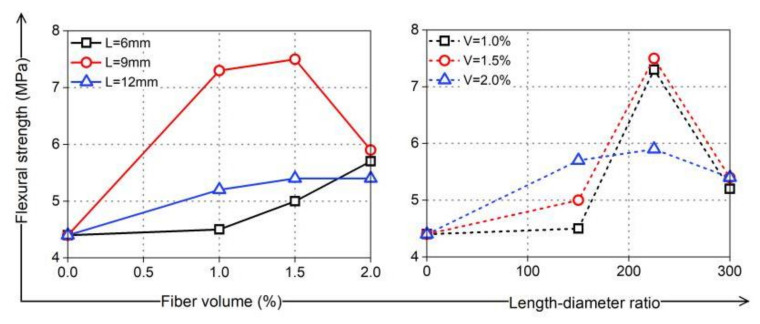
Flexural strength of PVA-reinforced coral cement-based composites.

**Figure 13 materials-16-02914-f013:**
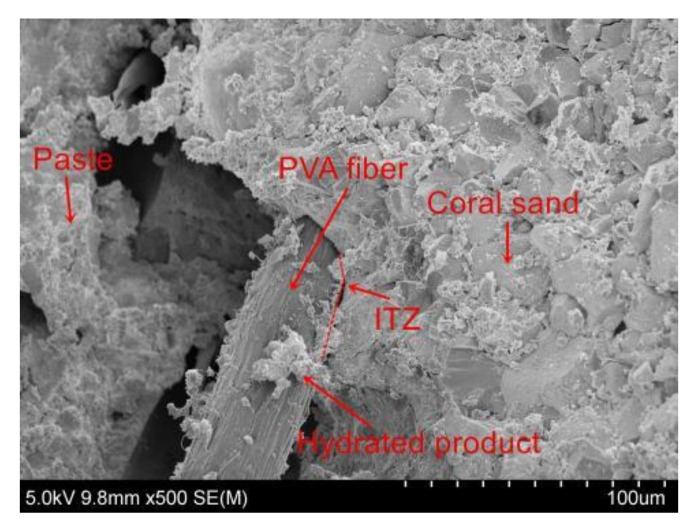
The interfacial transition zone between PVA fibers and the coral cement matrix.

**Figure 14 materials-16-02914-f014:**
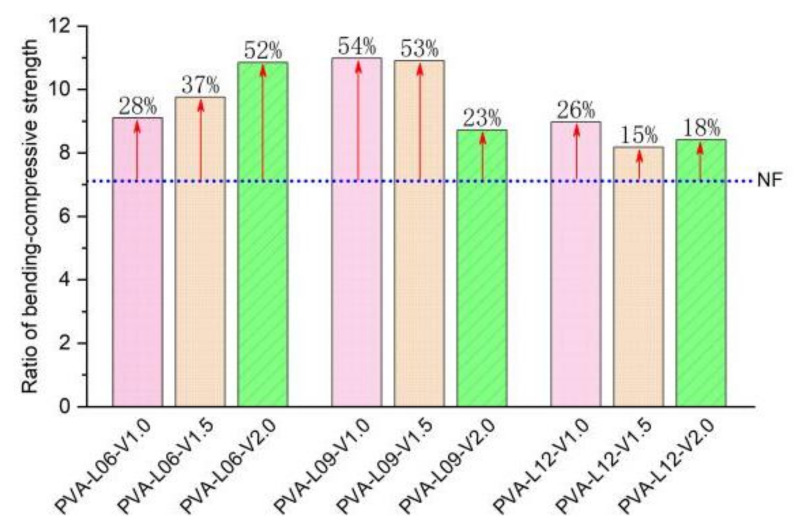
Relation between the growth rate of the bending-compressive strength ratio and L/D ratio.

**Figure 15 materials-16-02914-f015:**
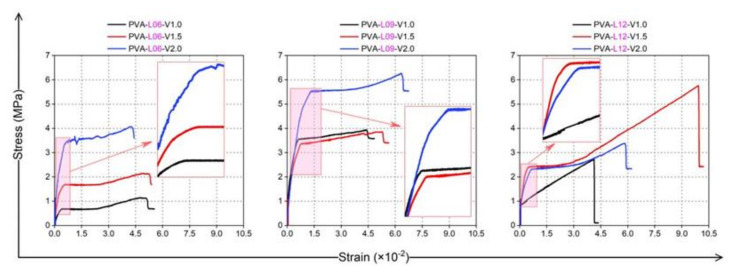
Tensile stress-strain curves of PVA fiber-reinforced coral cement-based composites specimens.

**Figure 16 materials-16-02914-f016:**
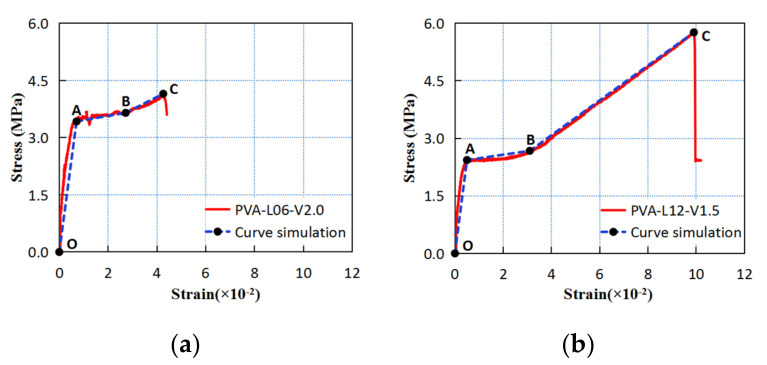
Comparison of the tensile stress-strain curves with the trilinear constitutive model curves. (**a**) PVA-L06-V2.0, (**b**) PVA-L12-V1.5.

**Figure 17 materials-16-02914-f017:**
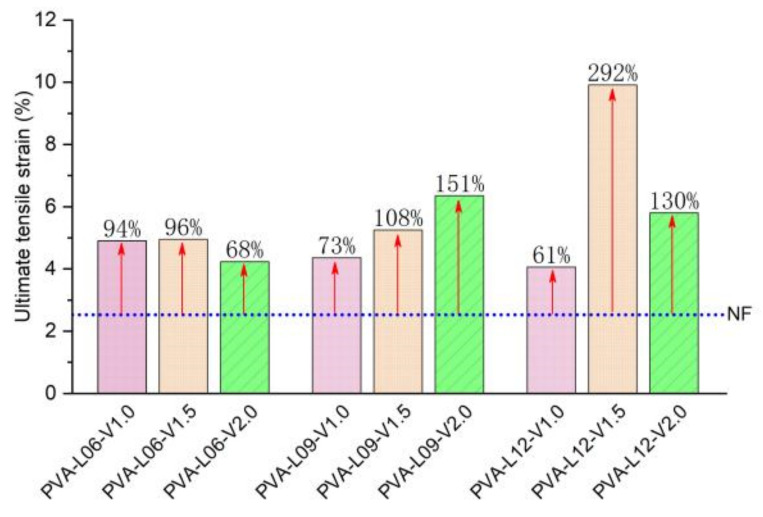
The growth rate of the ultimate tensile strain of PVA-reinforced coral cement-based composites.

**Figure 18 materials-16-02914-f018:**
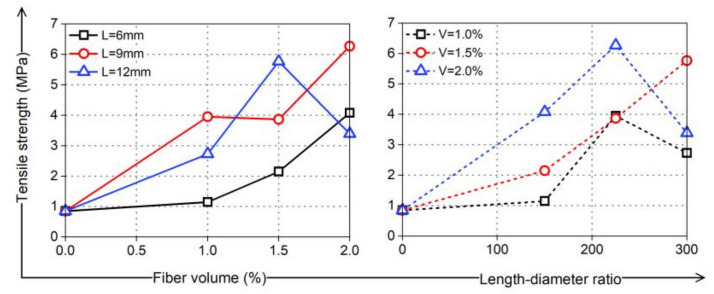
Tensile strength of PVA-reinforced coral cement-based composites.

**Table 1 materials-16-02914-t001:** The mechanical properties of the PVA fibers. (Data come from specifications).

Fiber Type	Diameter	Length	Density	Elongation at Break	Tensile Strength	Elastic Modulus
	(μm)	(mm)	(g/cm^3^)	(%)	(MPa)	(GPa)
PVA	40	6, 9, 12	1.3	17 ± 3.0	1400–1600	35–39

**Table 2 materials-16-02914-t002:** Mix proportions of PVA fiber-reinforced coral cement-based composites.

Mix ID	Coral Sand	Binder			Water	Admixture			Fiber Length	Fiber Volume
		Cement	Fly Ash	Quartz Powder		Water-Reducing Agent	Defoamer	Thickener		
	(kg/m^3^)								(mm)	(%)
NF	1350	787.5	112.5	112.5	270	3.6	4.0	3.15	-	-
PVA-L06-V1.0	1350	787.5	112.5	112.5	270	3.6	4.0	3.15	6	1.0
PVA-L06-V1.5	1350	787.5	112.5	112.5	270	3.6	4.0	3.15	6	1.5
PVA-L06-V2.0	1350	787.5	112.5	112.5	270	3.6	4.0	3.15	6	2.0
PVA-L09-V1.0	1350	787.5	112.5	112.5	270	3.6	4.0	3.15	9	1.0
PVA-L09-V1.5	1350	787.5	112.5	112.5	270	3.6	4.0	3.15	9	1.5
PVA-L09-V2.0	1350	787.5	112.5	112.5	270	3.6	4.0	3.15	9	2.0
PVA-L12-V1.0	1350	787.5	112.5	112.5	270	3.6	4.0	3.15	12	1.0
PVA-L12-V1.5	1350	787.5	112.5	112.5	270	3.6	4.0	3.15	12	1.5
PVA-L12-V2.0	1350	787.5	112.5	112.5	270	3.6	4.0	3.15	12	2.0

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
