# Peer review of "Experimental Study on Basic Mechanical Properties of PVA Fiber-Reinforced Coral Cement-Based Composites"

_materials, 2023, doi:10.3390/ma16072914_

Round 1

Reviewer 1 Report

Basic macroscopic mechanical properties of 9 different mix of PVA fiber-reinforced coral cement-based composites are considered in order to evaluate the strengthening mechanism of PVA fibers and select the optimal mix.

The results are interesting and can contribute to the development of new building materials, useful in particular situations.

Some suggestions could improve the paper:

All the experimental results should be reported in a table with also the mean value and the coefficient of variation, otherwise the results reported in the text lose value.

Pag. 3 line 100 in the sentance “After refining the particle size of coral sand and optimizing particle gradation and sieving …..” it would be useful to specify the criterion and references of optimizing particle gradation

It would be necessary to specify the type of bending test performed. Three points or four-point bending test?

As a starting point for future research, the determination of the fracture energy by bending test on notched specimens would be an interesting finding.

Author Response

Reviewer #1:

Comments and Suggestions:

Basic macroscopic mechanical properties of 9 different mix of PVA fiber-reinforced coral cement-based composites are considered in order to evaluate the strengthening mechanism of PVA fibers and select the optimal mix. The results are interesting and can contribute to the development of new building materials, useful in particular situations.

Some suggestions could improve the paper:

Point 1: All the experimental results should be reported in a table with also the mean value and the coefficient of variation, otherwise the results reported in the text lose value.

Response 1: Thank for the Reviewer’s very valuable comments. Concrete is a multiphase composite material composed of aggregate, mortar and interfacial transition zone. Its physical properties, such as elastic modulus and ultimate strength, have a certain randomness, as the Reviewer have commented. Therefore, we conducted three sets of parallel experiments for each working condition and took their average values to eliminate or weaken the influence of the randomness of concrete samples on the experimental results. This method has also been recognized and adopted by many scholars. While in order to show the regularity of the experimental results more directly, the experimental results reported in graph form rather than table and do not lose value.

Point 2:  Pag. 3 line 100 in the sentance “After refining the particle size of coral sand and optimizing particle gradation and sieving …..” it would be useful to specify the criterion and references of optimizing particle gradation

Response 2: Thank for the Reviewer’s rigorous and careful review of our manuscript. According to the comments of the reviewer, we have a supplement in the revised manuscript.

Point 3: It would be necessary to specify the type of bending test performed. Three points or four-point bending test?

Response 3: Thank for the Reviewer’s rigorous and careful review of our manuscript. According to the comments of the reviewer, we have a supplement in the revised manuscript.

Point 4: As a starting point for future research, the determination of the fracture energy by bending test on notched specimens would be an interesting finding.

Response 4: Thank for the Reviewer’s suggestions for our future research, and we will try to carry out the interesting experimental study in the near future.

Reviewer 2 Report

The authors investigated the use of PVA fibers with different lengths and different volumes as a reinforcement of composite concrete (based on corals aggregates. This research is classical and fits well with the scope of the journal. The following comments should be taken into account before publishing this manuscript:

Ø  General comments:

v 20 Figures are too many for a manuscript, so remove any unnecessary figures; Fig. 1, Fig. 4, Fig. 5-Fig. 9....

Ø  Specific comments:

1.     Introduction

·       Explain the objective/s and the novelty of your research in the last paragraph. B

2.      Line 98: “… its 98 main chemical component was calcium carbonate, as shown in Figure 1(a).”

·       Fig. 1(a) does not show anything related to the composition of the natural coral.

3.     Fig.1

·       Add a bar scale.

4.     Section 2.1

·       Add information about the source of the PVA fibres.

5.     Table 1

·       If these measurements are not your work, then add the source/reference.

6.     Line 122: “Their size is shown in 122 Figure 4(a).”

·       “Their dimensions” instead of “their size’’

·       Figure numbering should be in the order they appear in the text; Figure 3 cannot be after Figure 4.

7.     Line 203:’’ In general, as shown in Figure 11”

·       It seems Fig. 10 is missing?

8.     Line 226: “Therefore, the influence of PVA fibers on the coral cement-based composite compressive 226 strength was both positive and negative.”

·       According to Figure 11: I afraid that there is no significant change in the compressive strength, and these variations results from the technical errors. How many samples are used? Can you add he error bars?

9.     Line 238: “Therefore, the L/D 238 of PVA fibers was an important factor affecting the compressive strength reinforcement 239 effect of coral cement-based composites.”

·       The critical length of the fibres can be calculated theoretically.

10.  Section 4.2

·       Fig. 12 fits with your analysis except PVA-L12-V; the load-deflection curve is similar to that of the fibres, it seems the cement matrix was already broken.

Author Response

Comments and Suggestions:

The authors investigated the use of PVA fibers with different lengths and different volumes as a reinforcement of composite concrete (based on corals aggregates. This research is classical and fits well with the scope of the journal. The following comments should be taken into account before publishing this manuscript:

General comments:

Point 1: 20 Figures are too many for a manuscript, so remove any unnecessary figures; Fig. 1, Fig. 4, Fig. 5-Fig. 9…..

Response 1: Thank for the Reviewer’s rigorous and careful review of our manuscript. According to the comments of the reviewer, we have deleted Fig. 1 in the revised manuscript.

Since it is necessary for this paper to clearly display the shape of the special tensile specimens and the self-made fixture, as well as the failure modes characteristics of specimens after tests, FIG. 4 and FIG. 5-9 are retained.

Specific comments:

Point 1: Introduction

Explain the objective/s and the novelty of your research in the last paragraph. B

Response 1: Thank for the Reviewer’s rigorous and careful review of our manuscript. The objective/s and the novelty of our research had been explained in the introduction as “PVA fibers with high toughness, high crack resistance, high deformation and corrosion resistance…… are adopted to reduce the brittleness and improve the toughness of high-strength coral cement-based composites” and “change the current situation that coral cement-based composites can only meet the needs of civil use and some special cases”.

Point 2: Line 98: “… its 98 main chemical component was calcium carbonate, as shown in Figure 1(a).” Fig. 1(a) does not show anything related to the composition of the natural coral.

Response 2: Thank for the Reviewer’s rigorous and careful review of our manuscript. According to the “General comments” of the reviewer, Fig. 1 is an unnecessary figure, we have deleted Fig. 1 in the revised manuscript.

Point 3: Fig.1

Add a bar scale.

Response 3: Thank for the Reviewer’s rigorous and careful review of our manuscript. According to the “General comments” of the reviewer, Fig. 1 is an unnecessary figure, we have deleted Fig. 1 in the revised manuscript.

Point 4: Section 2.1

Add information about the source of the PVA fibres.

Response 4: Thank for the Reviewer’s rigorous and careful review of our manuscript. According to the comments of the reviewer, we have a supplement in the revised manuscript.

Point 5: Table 1

If these measurements are not your work, then add the source/reference.

Response 5: Thank for the Reviewer’s rigorous and careful review of our manuscript.These measurements in Table 1 are from the specifications for the PVA fibers. According to the comments of the reviewer, we have a supplement in the revised manuscript.

Point 6: Line 122: “Their size is shown in Figure 4(a).”

“Their dimensions” instead of “their size’’

Figure numbering should be in the order they appear in the text; Figure 3 cannot be after Figure 4.

Response 6: Thanks for the Reviewer’s rigorous and careful review of our manuscript. According to the comments of the reviewers, we have revised these mistakes in the revised manuscript.

Point 7: Line 203:’’ In general, as shown in Figure 11”

It seems Fig. 10 is missing?

Response 7: Thank for the Reviewer’s rigorous and careful review of our manuscript. According to the comments of the reviewer, we have renumbered the figures and revised this mistake in the revised manuscript.

Point 8: Line 226: “Therefore, the influence of PVA fibers on the coral cement-based composite compressive strength was both positive and negative.”

According to Figure 11: I afraid that there is no significant change in the compressive strength, and these variations results from the technical errors. How many samples are used? Can you add he error bars?

Response 8: As Reviewer pointed out “there is no significant change in the compressive strength”, this is due to the superimposition of two effects, One is a positive effect that “PVA fibers could restrain the transverse expansion of coral cement-based composites under unidirectional pressure …thus improving the compressive strength…”, the other one is a negative effect “PVA fibers …prevents air from being expelled from the matrix … caused an increase in the porosity in the coral sand in the hardened coral cement matrix and increased internal defects … reduce the matrix compressive strength and compactness.”

Three samples in each group of conditions are used in the tests. Actually, both prism compressive strength results and cubic compressive strength results correspond to the similar law showed in Figure 11 (Corresponding to Figure 9 in the revised manuscript), so the results are not from the technical errors probably.

Point 9: Line 238: “Therefore, the L/D of PVA fibers was an important factor affecting the compressive strength reinforcement effect of coral cement-based composites.”

The critical length of the fibres can be calculated theoretically.

Response 9: Thank for the Reviewer’s rigorous and careful review of our manuscript. Due to the particularity of coral aggregate and coral concrete, the strengthening mechanism of PVA fiber on coral cement-based composites is different from that of normal concrete. The traditional calculation methods for critical length of the fibres are not necessarily applicable to coral concrete. Calculate theoretically the critical length of the fibres is the content of our follow-up research, and the tests results of this paper can provide a basis for future theoretical research. Thank for the Reviewer’s suggestions again.

Point 10: Section 4.2

Fig. 12 fits with your analysis except PVA-L12-V; the load-deflection curve is similar to that of the fibres, it seems the cement matrix was already broken.

Response 10: Thank for the Reviewer’s rigorous and careful review of our manuscript. PVA-L12-V1.5 and PVA-L12-V2.0 also fit with the general trend of our analysis except PVA-L12-V1.0, the cement matrix had indeed already broken. This phenomenon may be caused by errors in the preparation process of the PVA-L12-V1.0 specimens. But, individual problems do not affect the overall analysis. According to the suggestions of reviewers, we have made an explanation in the revised manuscript.

Reviewer 3 Report

In this work, an experimental study is carried out to analyse mechanical properties of PVA fiber-reinforced coral cement-based composites, taking into account the fiber content and length-diameter ratio (L/D). PVA fibers are used in common cement-based composites to reduce the brittleness and improve the toughness of high-strength coral cement-based composites.

An improvement is clearly demonstrated in the paper when using PVA to obtain composites materials. However, I think paper needs some amelioration and I would suggest the following revision on the paper.

(1) The name of PVA should appears in the manuscript at least one time when the abbreviation PVA appeared in the first time.

(2) avoid long sentences for example sentence in lines 80 to 88

(3) In figure 17: the stress at the initial crack point (point A) for PVA-L12-V.. is lower than PVA-L09-V.. . Authors are invited to explain these results correctly.

(4) The fiber–matrix interfacial domain is a critical part of composites because load transfers from the matrix to the fiber and vice versa occur through the interface. Authors are invited to explain this point and what precautions are taken by authors to improve mechanical properties of PVA fiber-reinforced coral cement-based composites.

The reviewer recommends that the author do minor revision to the manuscript. Also, the reviewer would recommend that the author proofread the article thoroughly for typos and grammatical errors and avoid long sentences.

Author Response

Comments and Suggestions:

In this work, an experimental study is carried out to analyse mechanical properties of PVA fiber-reinforced coral cement-based composites, taking into account the fiber content and length-diameter ratio (L/D). PVA fibers are used in common cement-based composites to reduce the brittleness and improve the toughness of high-strength coral cement-based composites.An improvement is clearly demonstrated in the paper when using PVA to obtain composites materials. However, I think paper needs some amelioration and I would suggest the following revision on the paper.

Point 1: The name of PVA should appears in the manuscript at least one time when the abbreviation PVA appeared in the first time.

Response 1: Thank for the Reviewer’s rigorous and careful review of our manuscript. According to the comments of the reviewer, we have a supplement in the revised manuscript.

Point 2: avoid long sentences for example sentence in lines 80 to 88

Response 2: Thanks for the Reviewer’s rigorous and careful review of our manuscript. According to the suggestions of reviewers, we have made corresponding adjustments in the revised manuscript.

Point 3: In figure 17: the stress at the initial crack point (point A) for PVA-L12-V.. is lower than PVA-L09-V.. . Authors are invited to explain these results correctly.

Response 3: During tensile test process, the strengthening effect of PVA fibers gradually begins to emerge from the latter elastic stage. And the strengthening mechanism of PVA fibers is similar to that in the bending experiment. PVA fibers tend to curl when they are too long, and it is difficult for them to exert their tensile strain capacity when cracks occur in the bridge matrix, which limits the improvement of the coral cement-based composite ductility. So the stress at the initial crack point (point A) for PVA-L12-V.. is lower than PVA-L09-V.. Since the reasons have been explained in Section 4.2, there is no restatement in Section 4.3.

Point 4: The fiber–matrix interfacial domain is a critical part of composites because load transfers from the matrix to the fiber and vice versa occur through the interface. Authors are invited to explain this point and what precautions are taken by authors to improve mechanical properties of PVA fiber-reinforced coral cement-based composites.

Response 4: When cracks start to occur in the matrix, the load at the crack is transferred by the matrix to PVA fibers, and the fibers began to play a bridging role to transfer the load to the matrix without cracks. In order to improve mechanical properties of PVA fiber-reinforced coral cement-based composites, in the process of preparation, we added thickener and defoamer to improve the dispersibility of PVA fibers in matrix and reduce the defects caused by air bubbles.

Point 5: The reviewer recommends that the author do minor revision to the manuscript. Also, the reviewer would recommend that the author proofread the article thoroughly for typos and grammatical errors and avoid long sentences.

Response 5: Thanks for the Reviewer’s rigorous and careful review of our manuscript. According to the suggestions of reviewers, we have made corresponding adjustments in the revised manuscript.

Round 2

Reviewer 2 Report

The authors have addressed most of my concerns